# About the Transient Effects of Synthetic Amorphous Silica: An In Vitro Study on Macrophages

**DOI:** 10.3390/ijms24010220

**Published:** 2022-12-22

**Authors:** Anaëlle Torres, Véronique Collin-Faure, Daphna Fenel, Jacques-Aurélien Sergent, Thierry Rabilloud

**Affiliations:** 1Solvay/GBU Silica, 69003 Lyon, France; 2Chemistry and Biology of Metals, University Grenoble Alpes, CNRS UMR5249, CEA, IRIG-LCBM, 38054 Grenoble, France; 3Institut de Biologie Structurale, Université Grenoble Alpes, CEA, CNRS, 38000 Grenoble, France; 4Integrated Structural Biology Grenoble (ISBG) CNRS, CEA, Université Grenoble Alpes, EMBL, 38000 Grenoble, France; 5Solvay SA/Toxicological and Environmental Risk Assessment Unit, 1120 Brussels, Belgium

**Keywords:** persistence of the effects, macrophages, synthetic amorphous silica

## Abstract

Silica (either crystalline or amorphous) is widely used for different applications and its toxicological assessment depends on its characteristics and intended use. As sustained inflammation induced by crystalline silica is at the root of silicosis, investigating the inflammatory effects induced by amorphous silicas and their persistence is needed. For the development of new grades of synthetic amorphous silicas, it is also desirable to be able to understand better the factors underlying potential adverse effects. Therefore, we used an optimized in vitro macrophage system to investigate the effects of amorphous silicas, and their persistence. By using different amorphous silicas, we demonstrated that the main driver for the adverse effects is a low size of the overall particle/agglomerate; the second driver being a low size of the primary particle. We also demonstrated that the effects were transient. By using silicon dosage in cells, we showed that the transient effects are coupled with a decrease of intracellular silicon levels over time after exposure. To further investigate this phenomenon, a mild enzymatic cell lysis allowed us to show that amorphous silicas are degraded in macrophages over time, explaining the decrease in silicon content and thus the transiency of the effects of amorphous silicas on macrophages.

## 1. Introduction

Crystalline silica, which is widely represented in nature, is known to be the etiological agent of silicosis [1,2]. Silicosis occurs mainly from industrial activities such as mining, construction work, or sandblasting, but can also occur in rare cases from residential exposure to high levels of crystalline silica dust [3]. In order to alleviate these problems but also to widen the scope of the industrial use of silica, the chemical industry has introduced the production and use of other forms of silica, namely synthetic amorphous silica (SAS), which can be produced by a variety of routes, leading to products of different characteristics in terms of particle and aggregates sizes and structures. Because of the obvious parallel between the two forms of silica, extensive toxicological testing was carried out on synthetic amorphous silica (see Napierska et al. [4] or Fruijtier-Polloth [5] for review). In particular, comparative studies between crystalline and synthetic amorphous silica were carried out and concluded a strong effect persistence for crystalline silica, while the effects of amorphous silicas were shown to be transient [6,7,8,9] for all three major types of synthetic amorphous silicas, namely colloidal, precipitated, or pyrogenic. 

Opposed to other nanomaterials (e.g., silver nanoparticles), which are toxic for a wide range of cell types, the effects of silica, either crystalline or amorphous, seem to be mediated mainly via their effects on macrophages, for which silica is both toxic [10,11] and pro-inflammatory [12,13,14,15,16], while these effects are not found on other cell types [17]. Recent research has also shown that pyrolytic amorphous silica induces stronger responses than precipitated amorphous silica [18,19,20].

The selective toxicity of silica toward macrophages has attracted extensive research on its mechanisms [21,22,23,24], as the chemical pathways that might be at play are much less obvious than for other nanomaterials, e.g., those that can liberate toxic ions such as silver or zinc oxide. In comparison, the mechanisms by which synthetic amorphous silica induces only transient effects compared to crystalline silica have been much less studied, although it was shown that the silicon content of lungs in rats exposed to either amorphous or crystalline silica did not change at the same rate [25]. We thus decided to investigate this phenomenon in further detail by comparing how macrophages reacted to various grades of synthetic amorphous silica after exposure.

## 2. Results

### 2.1. Synthetic Amorphous Silicas Characterization 

The various silicas used in this study were characterized by transmission electron microscopy. The results are shown in Figure 1. As a nomenclature for this work, the first letter (S or M) indicated the size of the primary particles (small or medium), and the remaining letters (Pr, C, or Py) the type of SAS (respectively precipitated, colloidal or pyrogenic). Thus, for example, SPr is a precipitated silica with small primary particles, while MC is a colloidal silica with medium primary particles.

As expected, the colloidal silicas consisted of well dispersed nanoparticles, and the observed sizes corresponded to the specifications. The medium-particle colloidal silica (MC) consisted of particles of an average diameter of 29 ± 5 nm, while the small-particle colloidal silica (SC) consisted of particles of an average diameter of 10 ± 2 nm, as previously published [26]. The precipitated silicas consisted of micrometric or submicrometric aggregates of primary particles of 13 ± 11 nm (SPr) or 20 ± 6 nm (MPr), which formed in turn micrometric agglomerates. Finally, the pyrolytic silica also consisted of aggregates (micron range size) of primary particles close to 15–20 nm in size [26].

### 2.2. Viability Studies

In a first series of experiments, we determined which concentrations of the various synthetic amorphous silicas could be used on the macrophages, by performing viability assays. The results, shown in Figure 2, led us to the highest concentrations that could be used without inducing massive cell death. These concentrations ranged from 15–20 µg/mL for the colloidal silicas to 100 µg/mL for the precipitated silicas, the pyrogenic one being intermediate with a usable concentration of 35 µg/mL. 

Once these concentrations were determined, they were used in the uptake and functional experiments.

### 2.3. Mineralization Efficiency

As the mineralization efficiency may be different from one type of synthetic amorphous silica to another, we first determined this parameter in a medium that was close to a cellular lysate, i.e., a cell culture medium supplemented with horse serum. To this purpose, we mixed the synthetic amorphous silicas with the complete culture medium to the adequate concentrations, drew an aliquot portion that was submitted to the mineralization and silicon assay procedure, and added the remaining of the silica-supplemented medium to the cells. The results, shown in Table 1, indicate that the overall silicon determination yield varied from 20–25% for the precipitated silicas to 40–50% for the colloidal ones, the pyrogenic silica being once again in the middle.

These yields were used in the subsequent experiments to correct the silicon concentration measured in the cell lysates for these various efficiencies.

### 2.4. Silica Uptake by Cells

The cells were then incubated with the silica-containing media for 24 h. After this exposure period, the cell culture medium was removed, the cell layer was rinsed with culture medium without silica, then twice with a saline solution and then lysed directly in an adequate buffer. An aliquot of this cell extract suspension was then submitted to the mineralization and silicon assay procedure to determine the amount of silica that had been taken up by the cells, while another aliquot was used to determine the amount of cellular proteins in the extract, thereby affording a control related to the number of cells present in the culture well. The results are shown in Table 2. 

Because of the varied input silica concentrations, the amount of silica internalized by the cells was variable from one synthetic amorphous silica to another. Nevertheless, the proportion of silica internalized by the cells varied from, ca., 15% (pyrogenic silica) to 40% (medium-particle precipitated silica). When a statistical analysis (ANOVA + Tukey’s pairwise test) was performed, the uptake of the pyrogenic silica was found to be different from the uptake of all other silicas except for the small-particle precipitated silica (Appendix A). There was thus no correlation between the uptake and the size of the primary particle, nor with the overall size of the aggregates. 

### 2.5. Functional Effects of Silica on Cells

As silica is internalized by cells, we then studied the functional effects on macrophages, both immediately after exposure and after a recovery period. To this purpose, the cells were exposed to synthetic amorphous silicas for 24 h. The silica-containing medium was then removed and replaced by fresh complete culture medium without silica, and the cells were left to recover in this medium for 72 h, with one medium change at 36 h to avoid exhaustion and to keep the cells viable. As a first parameter, the phagocytic capacity of macrophages was studied through the uptake of fluorescent latex beads. The results, shown in Figure 3A, indicated a decrease in the percentage of phagocytic cells immediately after exposure for the three SAS composed of small primary particles, but not for the two SAS composed of larger primary particles. This effect was reversible at the end of the recovery period, indicating a transient effect only. However, a small but statistically significant decrease in the proportion of phagocytic cells was observed at the end of the recovery period for the colloidal SAS made of medium particles. Regarding the phagocytic activity of the phagocytic cells, as detected by the fluorescence intensity of the cells having internalized the fluorescent latex beads, the results shown on Figure 3B indicated a double effect for the colloidal SAS made of small particles. A decrease was observed immediately after exposure, followed by an increase at the end of the recovery period, indicating a “rebound effect”. 

We then tested the inflammatory response to SAS, by measuring the levels of interleukin-6 (IL-6) and tumor necrosis factor α (TNF-a) secreted immediately after exposure and at the end of the recovery period. We first tested the intrinsic inflammatory responses to SAS, by measuring the cytokine levels after treatment by the SAS alone. The results, shown in Figure 4A,B are very different for IL-6 and TNF-a. For Il-6, only the pyrogenic silica induced a significant increase in secretion, and this effect was persistent after the recovery period. These results were consistent with those described in the literature [27]. The situation was more complex for TNF-a. All SAS induced an increase in secretion immediately after exposure. At the end of the recovery period, the non-colloidal SAS showed a tendency to a return to normal values, although the actual values were still significantly higher than the control ones. Conversely, colloidal SAS showed persistence of the increase in TNF-a secretion. 

We then tested how the internalization of SAS may alter the inflammatory response to bacterial components, using LPS as a model compound. Here again the results, shown in Figure 4C,D, are different for IL-6 and TNF-a. For IL-6, a small but statistically significant increase was observed immediately after exposure for all SAS except for the pyrogenic one. However, this increase was reversible after the recovery period, except for the pyrogenic SAS, suggesting a delayed effect. For TNF-a, only the pyrogenic SAS and the colloidal one made of small particles showed an increase in secretion immediately after exposure. This effect was reversible at the end of the recovery period, except for the precipitated SAS made of medium size particles, for which a delayed increased was observed.

### 2.6. Persistence of Silica in Cells

As the transient nature of most responses may suggest a transient presence of silica in cells, the persistence of the silica in cells post-exposure was studied, both immediately after exposure and after the recovery period. The results in Table 3, show a significant decrease in the amount of synthetic amorphous silicas still present in the cells after the 72 h recovery period, with only one third to one half of the initial internalized dose still present in the cells after the recovery period. The lowest values were observed for the precipitated and colloidal silicas with small primary particles, while the highest value was observed for the pyrogenic silica.

A striking result was however observed for the medium-particle precipitated silica, where an apparent increase of the silicon concentration was observed at the end of the recovery period compared to the condition just after exposure. As the experimental scheme does not allow this to happen, the only remaining explanation possible was that the amorphous silica became more easily mineralizable in the cells during the recovery period, suggesting that the SAS were altered during the recovery period. 

### 2.7. Synthetic Amorphous Silica Observation after Four Days in Cells

In order to investigate the fate of SAS in cells, we exposed macrophages for 24 h to SAS and allowed them to recover without SAS. We then lysed the cells and imaged the silica by TEM. The results, shown in Figure 5, showed an aggregation of the colloidal SAS, but a strong disaggregation of precipitated and pyrogenic SAS. 

## 3. Discussion

With a yearly production (and use) in millions of tons worldwide, understanding the factors governing the biological effects of SAS is an important topic in the field of nanosafety, especially in the conceptual frame of crystalline silica-mediated silicosis and its toxicity. Considerable efforts have been devoted to unravelling the structural bases of silica toxicity [21,22,23,24], but their application in the field of SAS is not straightforward. For colloidal silicas the influence of the particle size was demonstrated [28,29]. For more aggregated silicas such as precipitated or pyrogenic silicas, which represent the highest part of SAS produced, a “divide and conquer” strategy was used to demonstrate that small aggregates were more toxic than large ones [30]. This latter approach is however not devoid of issues, as the fracturation process used to disrupt the large aggregates may produce the reactive entities that cause the silica adverse effects [23,24]. This is why we rather chose to compare different commercial SAS with different parameters to gain insights into the determinants of SAS adverse effects. It appears that the principal driver of toxicity is the overall particle/agglomerate size, with the colloidal silicas being more toxic than the more aggregated ones. The difference, however, was less than one order of magnitude, showing that all SAS exhibit a somewhat similar toxicity. When going into more details into the inflammatory effects of SAS on macrophages, the second important driver was the specific surface area. The SAS with the lowest specific surface area (55 m^2^/g) was both the least toxic and the one with the lowest effects, even when used at a higher concentration than the other ones. Quite remarkably, the adverse effects did not correlate well with the amount of SAS internalized, suggesting here again an influence of the overall particle/agglomerate size and of structural details [24], as the pyrogenic SAS showed, for example, more persistent pro-inflammatory effects, consistent with reports indicating severer effects with this type of silica [19,20]. However, the adverse effects of the other SAS on macrophages were mostly transient, in contrast with the persistent effects described both in vivo [7,8,9] and in the same in vitro macrophage system for crystalline silica [31].

When pursuing the goal of “safe by design” SAS, persistence of the effects is one of the key points to consider. While sustained inflammation induced by crystalline silica is at the root of silicosis, such a process has not been shown to occur with SAS. In vivo studies have shown that the pulmonary effects of SAS are transient [6,7,8], and this is correlated with a decrease of the amount of silica (measured through elemental Si) in the lungs [9]. Moreover, reports showing persistent effects of pyrogenic silica showed that a repetitive dosing was needed to induce persistent effects [20], suggesting that counteracting the SAS elimination process was required to induce the adverse effects. Indeed, when a true repeated daily exposure was used on macrophages, a low silicon accumulation was observed [32], in line with the elimination mechanism shown here.

However, two widely different hypotheses can explain the SAS elimination process. Once the silica has been captured by the macrophages, either the particle-laden macrophages are eliminated as a whole, as described for TiO_2_ particles [33], or the silica is dissolved in the macrophages and then eliminated, e.g., as silicate. To test the second hypothesis, we measured the silicon levels in macrophages exposed to SAS immediately after exposure and after a 72 h post-exposure recovery period. Silicon elimination was observed confirming this hypothesis. Furthermore, we were able to observe a disaggregation of precipitated and pyrogenic SAS in macrophages at the end of this recovery period, while colloidal silicas, which are originally made of isolated primary particles, were aggregated by the macrophages, a paradoxical behavior that certainly deserves further investigation. The electron microscopy images of the silica ages in cells do not allow the determination of whether colloidal SAS really becomes aggregated in cells upon ageing or if the particles are just compressed in the phagosomes and that the mild lysis process used here does not just disperse them.

Overall, these results strongly argue for an intracellular dissolution process after silica uptake by the macrophages as the SAS elimination process. As the macrophage phagolysosome is known to have an acid pH [34] and as silica is known to be stable under acidic conditions, this dissolution appears puzzling at first glance. However, it has been shown recently that the phagosome pH in macrophages having internalized silica is rather close to neutral [35]. Moreover, it is known that different polarizations of macrophages induce different pHs in the phagosome, with M2 macrophages showing acidic phagosomes and M1 macrophages showing more neutral phagosomes [36]. As the internalization of silica induces the production of pro-inflammatory cytokines such as IL-6 and TNF-a, as shown here, and also the expression of CD38 [32], an M1 polarization marker [37], there is clear evidence that macrophages having internalized silica are skewed towards the M1 state and are thus likely to have non-acidic lysosomes with a high content of oxidizing species. This peculiar environment, together with the presence of proteins in the phagolysosome, which complex a wide variety of ions, may favor the silica degradation observed here. 

This further shows the variety of the particle elimination processes, which can range from particle dissolution (as shown here or for metal-based nanoparticles [38,39]) to macrophage-mediated elimination, either direct [33] or indirect [40], or translocation through the epithelia into secondary organs [41]. Although simple in vitro systems can investigate easily only mechanisms linked to one cell type, they can do it in detail, as shown here in the investigation of the short effect persistence for SAS. 

## 4. Materials and Methods

### 4.1. SAS Observation and Characterization

In this work, five different synthetic amorphous silicas (SAS) were used:(i)a colloidal silica with small (10 nm) particles, purchased from Sigma-Aldrich (#420808, batch #MKBH8895V, St. Louis, MI, USA), designated as SC—this silica shows a specific area of 220 m^2^/g;(ii)a colloidal silica with medium (29 nm) particles, purchased from Sigma-Aldrich (#420778, batch #MKCK1767), designated as MC—this silica shows a specific area of 140 m^2^/g;(iii)a precipitated silica with small (13 nm) primary particles, from Solvay Silica (batch #F9C8C4034) designated as SPr—this silica shows a specific area of 220 m^2^/g;(iv)a precipitated silica with medium (20 nm) primary particles, from Solvay Silica (batch #F9I21CL032) designated as MPr—this silica shows a specific area of 55 m^2^/g;(v)a pyrogenic silica with small (15–20 nm) primary particles, purchased from VWR (#ACRO403731500, batch# A0403808), designated as SPy—this silica shows a specific area of 200 m^2^/g.

For material characterization with TEM microscopy, samples were diluted to 100 µg/mL. For negative stain on grid technique (SOG), 10 µL was added to a glow discharge grid coated with a carbon supporting film for 5 min. The excess solution was soaked off by a filter paper and the grid was air-dried. The images were taken under low dose conditions (<10 e^−^/Å^2^) with defocus values between 1.2 and 2.5 μm on a Tecnai 12 LaB6 electron microscope at 120 kV accelerating voltage using CCD Camera Gatan Orius 1000.

### 4.2. Nanoparticles

The SAS were suspended in water at 10 mg/mL, they were sonicated in an ultrasonic bath for 10 min and then they were incubated overnight at 80 °C in a preheated water bath. SAS were sonicated again for 10 min before use.

### 4.3. Cell Culture

The murine macrophage cell line J774A1, obtained from the European Cell Culture Collection (Salisbury, UK), was used for the cellular internalization studies. The cells were routinely grown on non-adherent plastic vessels in DMEM supplemented with 10% fetal bovine serum (FBS) and transferred to adherent 6-well plates for exposure to silica, as described below. Cells were used at passage numbers from 5 to 15 post-reception from the repository.

For treatment with SAS, cells were seeded at 300,000 cells/mL in DMEM + 1% horse serum (HS) and left for 72 h at 37 °C for cell adhesion and for growth at confluence. The cells were then treated using the following scheme where the control cells were not exposed to SAS. The recovery scenario consists of an acute exposure (24 h) followed by a 72 h period without any SAS. The acutely exposed cells were exposed to SAS for 24 h for the same time as the last 24 h of recovery of the secondary scenario cells (in order to have cells with the same ageing on the plates) [31]. In this study, we used 15 µg/mL for colloidal silica 140 m^2^/g (MC), 20 µg/mL for colloidal silica 220 m^2^/g (SC), 50 µg/mL for small-particle precipitated silica (SPr), 100 µg/mL for medium-particle precipitated silica (MPr) and for corundum (Cor). Except for corundum, these doses corresponded to the LD20 of the various forms of SAS, previously determined in preliminary experiments.

### 4.4. Viability Assay

The lethal dose 20 was determined for each SAS. The cells were seeded in 6-well culture plates at 600,000 cells/mL in DMEM FBS 10% and left for 24 h at 37 °C with 5% CO_2_ for cell adhesion. Then, cells were exposed for 24 h to the indicated doses of SAS. The cells were harvested in phosphate buffer saline (PBS 1X), and centrifuged for 5 min at 200× *g*. The pellets were resuspended in PBS 1X with 1 µg/mL propidium iodide (excitation 533 nm, and emission 617 nm; Merck P4864). Cells were analyzed with FacsCalibur flow cytometer by using CellQuestPro software 6.0, Becton Dickinson, Le Pont-de-Claix, France.

The control experiments, which were intended to take into account potential artifacts, are described in the Appendix A. 

### 4.5. Phagocytic Activity Assay

This experiment allows evaluation of the phagocytic capacity of the cells. Indeed, the fluorescent latex beads have a diameter of 1 µm, which corresponds to the size of bacteria. Briefly, after the exposure (acute or recovery period), the cells were exposed for 3 h to FITC-latex beads (previously coated in an FBS 10%–PBS 1X solution for 30 min at 37 °C, Sigma L4655, excitation 492 nm, emission 517 nm). Then cells were harvested and washed with PBS, centrifuged 5 min at 1200 rpm, the pellets were suspended with 3 mL of water for a few seconds and 1 mL of NaCl (3.5%) was added under vortex mixing to restore the osmotic pressure. The cells were centrifuged, and the pellets were resuspended with 200 µL of PBS 1X–propidium iodide 1 µg/mL, and analyzed by flow cytometry.

The control experiments, which were intended to take into account potential artifacts, are described in the Appendix A

### 4.6. Cytokine Secretion Dosages

As previously described, cells were grown in 6-well plates and exposed to the indicated dose of silica particles for 24 h (followed or not by a recovery period) at 37 °C. The cells were also primed or not with LPS (100 ng/mL) for the last 18 h of culture. The supernatants were collected, centrifuged (200× *g*, 5 min) to eliminate non adherent cells, and frozen at −20 °C until use. The cytokine dosage was carried out with the Cytometric Bead Array Mouse Inflammation Kit (BD Biosciences), and analyzed with FCAP Array software (3.0, BD Biosciences, Le Pont-de-Claix, France). This Flex-set kit allows measuring Interleukin-6 (IL-6) and TNF protein levels in a single sample. The mixed capture beads were added to all assay tubes containing supernatant samples and standards (from 0 to 5000 pg/mL), the mouse inflammation phycoerythrin (PE, excitation 488 nm, emission 575 nm) detection reagent was added, and the mixture was incubated for 2 h at room temperature, protected from light. The wash buffer was added to each tube, which were then centrifuged 5 min at 200× *g*, the pellets were resuspended with the wash buffer, and analyzed by FacsCalibur flow cytometer.

The control experiments, which were intended to take into account potential artifacts, are described in the Appendix A.

### 4.7. SAS Quantification by ICP-AES 

#### 4.7.1. Sample Preparation

For SAS quantification in water or medium (DMEM HS 1%), the SAS were added in T25 flasks at a final concentration of 100 µg/mL. The first sample was taken at Day 0 and allowed the mineralization efficiency for each SAS to be determined. The flasks were then incubated for four days at 37 °C, and then, the second sample was taken.

For SAS internalization and quantification in the cells, the cells were grown on adherent 6-well plates in DMEM supplemented with 1% horse serum (DMEM HS 1%). The scheme of exposure has already been described in [27,31]. The acutely exposed cells were exposed for 24 h and lysed at the end of the exposure. The recovery cells were exposed for 24 h, in parallel to the acutely exposed ones, and this exposure was followed by a 72 h recovery period without any SAS and with one medium change. Then, the cells were lysed as the acutely exposed cells. The lysis solution contained 5mM HEPES pH 7.5, 0.75 mM spermine tetrahydrochloride, and 0.1% SB 3–14. One milliliter of lysis solution was added in the wells, after the medium had been removed and the cell layer washed once with PBS. After quantification of proteins in each lysate with Bradford assay [42], the lysates were aliquoted and frozen. The day before the quantification by ICP-AES, the samples were mineralized: in each sample, an equal volume of 1N potassium hydroxide was added, the samples were then incubated overnight at 80 °C in a preheated water bath [43,44].

#### 4.7.2. ICP-AES Dosage

The standards were first prepared, in HNO_3_ 10%; the standard was purchased from Sigma-Aldrich (ref 92091). At the end of the standard measurement, samples were prepared two at a time (base-mineralized silica is not stable in HNO_3_): from 125 µL to 500 µL of mineralizate was added to *qsp* 6 mL of HNO_3_ 10%. The samples were measured in an ICP-AES Shimadzu 9000 with ICPE solution Launcher software (v1.31, Shimadzu France, Noisiel, France).

### 4.8. TEM Analysis of Internalized Silica

For the TEM analysis of internalized SAS by macrophages, cells were exposed to SAS as described above. After the recovery scenario or the acute exposure, the cells were harvested in PBS 1X and centrifuged. The pellets were lysed in HEPES buffer 10 mM, MgCl_2_ 2 mM (10 volume of lysis solution for 1 volume of cell pellet). Serratia marcescens nuclease (Benzonase^®^, ref 71206-3 from Sigma-Aldrich) was added at a concentration of 5 U/mL, and the cells were incubated at 37 °C under stirring, for 6 h. Then, a solution containing Proteinase K (ref P4850 Sigma) and CaCl_2_ was added (final concentrations 100 µg/mL and 2 mM respectively). The cells were incubated overnight. Then, the solutions were stocked at 4 °C until their use in TEM, the same day. These samples were analyzed by TEM with the same parameters as described above.

## Figures and Tables

**Figure 1 ijms-24-00220-f001:**
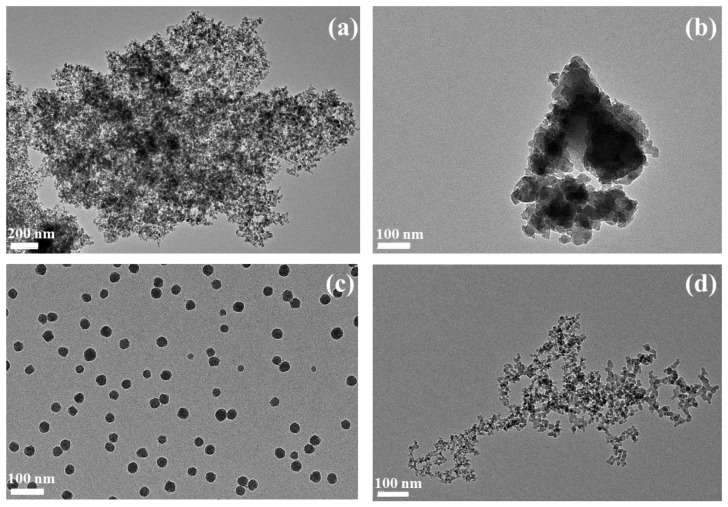
SAS characterization by transmission electron microscopy at Day 0 in water. (**a**) SPr 220 m^2^/g 23,000×. (**b**) MPr 55 m^2^/g 23,000×. (**c**) MC 140 m^2^/g 23,000×. (**d**) SPy 200 m^2^/g 23,000×.

**Figure 2 ijms-24-00220-f002:**
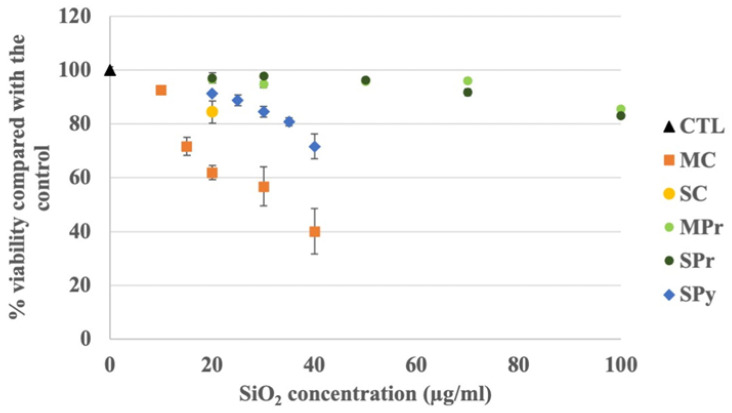
Viability assays of SAS exposure (24 h), on J774A.1 macrophages. Precipitated silicas are in green, colloidal silicas in orange, and the pyrogenic SAS in blue. Round dots are for small particle SAS and square dots for medium particle SAS; CTL: control, untreated cells; MC: medium-particle colloidal silica; SC: small-particle colloidal silica; SPr: precipitated silica with small primary particles; MPr: precipitated silica with medium-size primary particles; SPy: pyrogenic silica with small primary particles. The lethal doses 20 were estimated at 15 µg/mL for MC, 20 µg/mL for SC, 100 µg/mL for SPr and MPr, and 35 µg/mL for SPy.

**Figure 3 ijms-24-00220-f003:**
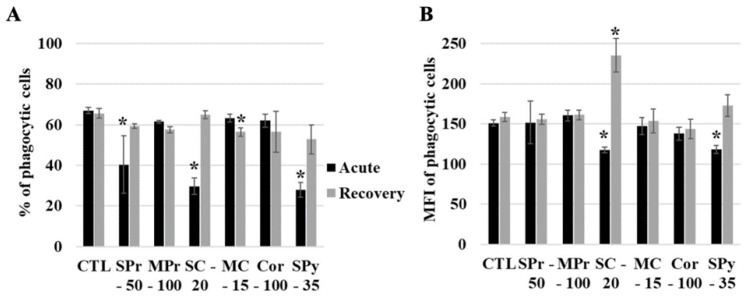
Phagocytic activity of cells exposed to SiO_2_ particles. The acute exposure is (or is not) followed by a 72-h recovery period. CTL: control cells, SPr: small-particle precipitated silica 220 m^2^/g, MPr: medium-particle precipitated silica 55 m^2^/g, SC: colloidal silica 220 m^2^/g, MC: colloidal silica 140 m^2^/g, Cor: corundum, SPy: small-particle pyrogenic silica 200 m^2^/g. Black bars for acutely exposed cells, grey bars for recovery protocol. The data are expressed as mean ± standard deviation. (**A**) Proportion of phagocytic cells. (**B**) Phagocytic activity of the cells. Statistical significance in a Mann-Whitney test: * *p* < 0.05.

**Figure 4 ijms-24-00220-f004:**
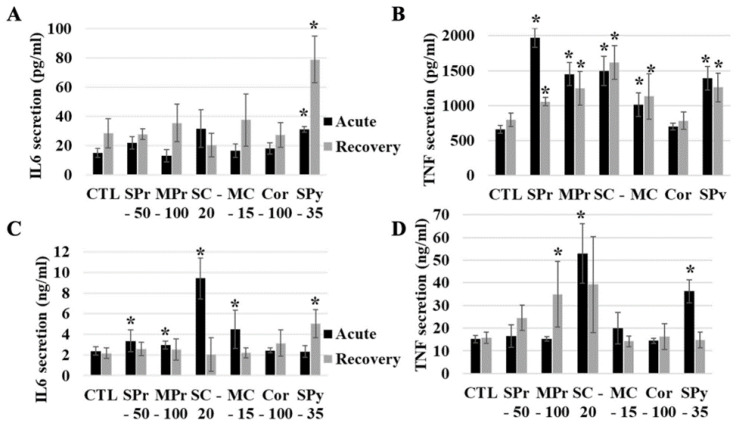
Cytokine responses of cells exposed to SiO_2_ particles and stimulated (or not) with lipopolysaccharide. The acute exposure is (or is not) followed by a 72-h recovery period. CTL: control cells, SPr: small-particle precipitated silica 220 m^2^/g, MPr: medium-particle precipitated silica 55 m^2^/g, SC: colloidal 220 m^2^/g, MC: colloidal 140 m^2^/g, Cor: corundum, SPy: small-particle pyrogenic silica 200 m^2^/g. Black bars for acutely exposed cells, grey bars for recovery protocol. The data are expressed as mean ± standard deviation. IL-6: interleukin 6; TNF: tumor necrosis factor alpha. (**A**) Intrinsic IL-6 secretion of cells exposed to SiO_2_. (**B**) Intrinsic TNF secretion. (**C**) IL-6 secretion after LPS stimulation. (**D**) TNF secretion after LPS stimulation. Statistical significance in a Mann-Whitney test: * *p* < 0.05.

**Figure 5 ijms-24-00220-f005:**
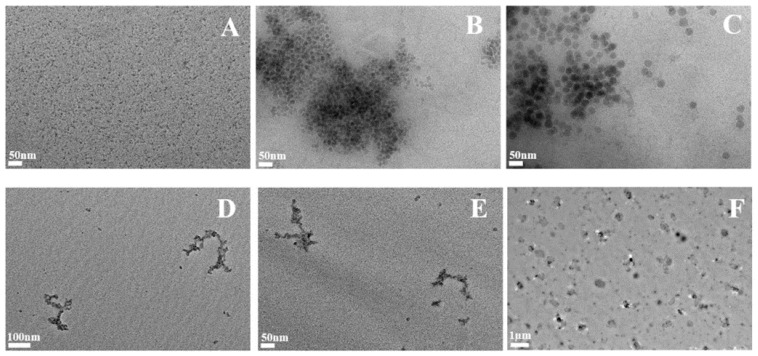
Examination by TEM of SAS extracted from cells. Macrophage cells were exposed to SAS for 24 h and allowed to recover for 72 h. At the end of the culture the cells were recovered, lysed and the lysate digested sequentially by a nuclease and a protease. The clarified lysate was examined by TEM to investigate the state of the internalized SAS. (**A**) control cells (no silica treatment) magnification 30,000×. (**B**) SC: small-particle colloidal silica 220 m^2^/g 30,000×. (**C**) MC: medium-particle colloidal silica 140 m^2^/g 30,000×. (**D**) SPr: small-particle precipitated silica 220 m^2^/g 23,000×. (**E**) MPr: medium-particle precipitated silica 55 m^2^/g 23,000×. (**F**) SPy: small-particle pyrogenic silica 200 m^2^/g 23,000×.

**Table 1 ijms-24-00220-t001:** Mineralization efficiency according to synthetic amorphous silica forms.

Condition	Amount Measured in Well (µg)	Measured Amount/Well (µg) (Mean ± SD)	Mineralization Yield	Mean Mineralization Yield
CTL	1.531	0.988 ± 0.496		
	0.558			
	0.876			
SC—20	21.760	16.213 ± 4.918	0.544	0.405
	12.384		0.310	
	14.496		0.362	
MC—15	14.560	14.357 ± 0.526	0.485	0.479
	14.752		0.492	
	13.760		0.459	
SPr—100	54.368	53.461 ± 2.939	0.272	0.267
	50.176		0.251	
	55.840		0.279	
MPr—100	29.152	37.92 ± 9.267	0.146	0.190
	36.992		0.185	
	47.616		0.238	
SPy—35	28.480	27.563 ± 1.989	0.407	0.394
	25.280		0.361	
	28.928		0.413	

**Table 2 ijms-24-00220-t002:** Determination of silica internalized in cells after acute exposure.

Cell Lysate (Acute)						
Condition	Amount Measured in Well	Corrected Amount/Well *	Amount/Well (Mean ± SD)	ng Si/µg Prot	ng Si/µg Prot (Mean ± SD)	Fraction Silica Incorporated in Cells
CTL	0.208	0.21	0.27 ± 0.30	0.01	0.013 ± 0.01	N/A
	0.598	0.60		0.03		
	0.018	0.02		0.00		
SC—20	6.944	12.76	13.95 ± 1.08	9.78	10.09 ± 0.94	0.35
	4.603	14.87		11.15		
	5.157	14.23		9.34		
MC—15	6.368	13.12	12.16 ± 1.72	8.07	6.67 ± 1.38	0.41
	5.003	10.17		5.31		
	6.050	13.19		6.64		
SPr—100	13.328	49.03	43.65 ± 11.09	38.61	41.74 ± 3.02	0.22
	12.800	51.02		44.63		
	8.624	30.89		41.98		
MPr—100	13.440	92.21	85.04 ± 14.81	66.98	55.94 ± 13.40	0.43
	17.552	94.90		59.82		
	16.192	68.01		41.03		
SPy—35	5.864	14.41	12.18 ± 4.95	8.86	6.77 ± 3.04	0.17
	5.643	15.63		8.16		
	2.690	6.51		3.28		

* The yield-corrected amount is calculated by dividing the measured amount by the mineralization yield determined in Table 1.

**Table 3 ijms-24-00220-t003:** Determination of the internalized silica in the cells, and still present after the recovery period.

Cell Lysate (Recovery)						
Condition	Amount Measured in Well	Yield Corrected Amount/Well *	Amount/Well (Mean ± SD)	ngSi/µg Protein (Mean ± SD)	Fraction Silica Incorporated in Cells	Ratio D4/D1
CTL	0.831	0.831	0.63 ± 0.45	0.31 ± 0.22		
	0.947	0.947				
	0.118	0.118				
SCc20	5.051	13.444	4.89 ± 7.43	2.09 ± 3.06	0.12	0.35
	0.003	0.009				
	0.456	1.214				
MC—15	3.662	9.747	5.34 ± 3.99	3.70 ± 3.01	0.18	0.44
	1.616	4.301				
	0.745	1.984				
SPr—100	5.418	14.419	13.18 ± 1.60	5.47 ± 0.99	0.07	0.30
	4.275	11.378				
	5.162	13.737				
MPr—100	42.592	113.357	97.36 ± 15.74	44.29 ± 7.90	0.49	1.14
	30.768	81.888				
	36.384	96.835				
SPy—35	2.270	6.043	5.84 ± 1.87	2.88 ± 0.61	0.08	0.48
	2.859	7.610				
	1.457	3.878				

* The yield-corrected amount is calculated by dividing the measured amount by the mineralization yield determined in Table 1.

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
