# Peer review of "About the Transient Effects of Synthetic Amorphous Silica: An In Vitro Study on Macrophages"

_ijms, 2022, doi:10.3390/ijms24010220_

Round 1
Reviewer 1 Report
Although the research in this manuscript is attractive and well-designed, I don't believe that the present format is appropriate for publishing.
Author Response
Reply to reviewer 1
Comment
Although the research in this manuscript is attractive and well-designed, I don't believe that the present format is appropriate for publishing.
Reply
As we cannot make anything of this comment in a revision, we will focus on the other reviewers’ comments
Reviewer 2 Report
figure 1 and figure 5, these figures have different scales, are difficult to compare between each other.
The order of the methods in material and methods should be changed
suggestion:
4.1. SAS observation and characterization
4.2. Nanoparticles
4.3. Cell culture
4.4. Viability assay
the rest are ok.
Author Response
Reply to reviewer 2
Comments and Suggestions for Authors
figure 1 and figure 5, these figures have different scales, are difficult to compare between each other.
Reply.
This is due to the fact that the objects are different in size, so that the scale is adapted to their best visualization. Thus we prefer to keep the initial scales for this reason
Comments and Suggestions for Authors
The order of the methods in material and methods should be changed
suggestion:
4.1. SAS observation and characterization
4.2. Nanoparticles
4.3. Cell culture
4.4. Viability assay
the rest are ok.
Reply.
Thank you for this suggestion. The order has been modified as requested
Reviewer 3 Report
This paper describes a thorough in vitro investigation of synthetic amorphous silica particle toxicity using macrophages. This is a valuable study topic, but the authors have not conducted sufficient control experiments to rule out biases and artifacts.
Sections 4.2, 4.5 – The authors need to evaluate for the potential of particles to bias the flow cytometer measurements as described in these publications (https://pubs.acs.org/doi/abs/10.1021/acs.chemrestox.9b00165; https://doi.org/10.1016/j.colsurfb.2018.09.021). Also, a positive control is needed. The authors need to test for the potential leaching of fluorophores from the latex beads in the presence of cells and the potential for biases from that.
Section 4.7 - Control experiments are needed for the cytokine experiments to rule out biases and artifacts as described in depth in this paper (https://pubs.acs.org/doi/abs/10.1021/acs.chemrestox.9b00165).
Section 4.8.2 – The authors need to analyze the recovery of the digestion and analysis process potentially using a reference material if possible.
Section 4.8.1 –Doesn’t mineralization imply that only the dissolved Si is measured? If yes, how were particles differentiated from dissolved Si using this procedure? Perhaps the authors need to clarify what exactly they mean by mineralization if my interpretation is wrong.
Additional comments
Section 2.1, 2.7 – The authors need to provide information such as a size distribution, not only the mean particle size.
Figure 2 – Error bars are needed. Describe what the abbreviations in the figure key are in the figure caption.
Tables 1,2, 3 – I recommend adding standard deviation values. If desired to simplify these tables, the raw data could be moved to the supplemental information with only the summary results in the tables.
Line 114 – What is an “adequate buffer”? I suggest to provide additional details.
Lines 122-129 – Results from the statistical analysis should be added to the table.
Figure 3 – What are the error bars (e.g., standard deviation values)?
Section 4.6 – Where are the results from this assay?
Author Response
Reviewer 3
Thank you very much for your report. We have done our best to address the concerns risen on the initial manuscript, and the additions appear in purple in the revised version.
Comment
This paper describes a thorough in vitro investigation of synthetic amorphous silica particle toxicity using macrophages. This is a valuable study topic, but the authors have not conducted sufficient control experiments to rule out biases and artifacts.
Comment
Sections 4.2, 4.5 – The authors need to evaluate for the potential of particles to bias the flow cytometer measurements as described in these publications (https://pubs.acs.org/doi/abs/10.1021/acs.chemrestox.9b00165; https://doi.org/10.1016/j.colsurfb.2018.09.021).
Also, a positive control is needed.
Reply :
we have carried out additional control experiments, which are now compiled in a supplementary material section. The control experiments for this precise point is now control experiment 1
Comment
The authors need to test for the potential leaching of fluorophores from the latex beads in the presence of cells and the potential for biases from that.
Reply :
we have carried out additional control experiments, which are now compiled in a supplementary material section. The control experiments for this precise point is now control experiments 2 & 3
Comment
Section 4.7 - Control experiments are needed for the cytokine experiments to rule out biases and artifacts as described in depth in this paper (https://pubs.acs.org/doi/abs/10.1021/acs.chemrestox.9b00165).
Reply :
we have carried out additional control experiments, which are now compiled in a supplementary material section. The control experiments for this precise point is now control experiments 4
Comment
Section 4.8.2 – The authors need to analyze the recovery of the digestion and analysis process potentially using a reference material if possible.
Reply :
As the purpose of these experiments is purely descriptive, the recovery is not a real concern and we will not address this point.
Comment
Section 4.8.1 –Doesn’t mineralization imply that only the dissolved Si is measured? If yes, how were particles differentiated from dissolved Si using this procedure? Perhaps the authors need to clarify what exactly they mean by mineralization if my interpretation is wrong.
Reply :
When elemental analyses are carried out, the name of the game is to bring as much as possible of the element of interest in the analysis. This is especially true when the element of interest is present in the form of complexes or of particles, both of which can savenge the element from the analysis.
Thus, the purpose of a mineralization procedure is to dissolve the element of interest into a soluble form that can be analyzed further. In an ideal world, the yield of the mineralization process should be 100 %. This is unfortunately not the case in some occasions, so that teh mineralization yield must always be known. This is exactly the purpose of the experiments described in Table 1, when a completely know amount of each silica is processed via mineralization, showing that the end results differ from one form of silica to another. This yieldis then used in subsequent experiments to compensate for the variable efficiency of the process (e.g. in Table 2). In order to differentiate soluble Si from particulate one, we used a centrifugation process. However, being in the supernatant does not necessarily mean that it is soluble Si, as very small silica nanoparticles may not pellet during the centrifugation. This means that what qualification between soluble Si and particulate Si is very difficult to carry out, explaining why we are so cautious in the interpretation of our results.
Additional comments
Comment
Section 2.1, 2.7 – The authors need to provide information such as a size distribution, not only the mean particle size.
Reply
done, in the results section
Comment
Figure 2 – Error bars are needed. Describe what the abbreviations in the figure key are in the figure caption.
Reply
done, in the results section
Comment
Tables 1,2, 3 – I recommend adding standard deviation values. If desired to simplify these tables, the raw data could be moved to the supplemental information with only the summary results in the tables.
Reply
done
Comment
Line 114 – What is an “adequate buffer”? I suggest to provide additional details.
Reply
this awkward wording has been removed and replaced by lysis buffer, which is defined in the material and methods section
Comment
Lines 122-129 – Results from the statistical analysis should be added to the table.
Reply
done, mentioned in supplementary Table 1
Comment
Figure 3 – What are the error bars (e.g., standard deviation values)?
Reply
done, mentioned in figure captions
Comment
Section 4.6 – Where are the results from this assay?
Reply
Thank you for pointing out this mistake, section removed as we did not mention any results from NO production
Round 2
Reviewer 3 Report
The paper should be acceptable after very minor revisions. First, the names of the control experiments in the Supplemental Material need to be revised (e.g., Experiment AA). Second, the authors have not added uncertainty values to the tables as suggested.
Author Response
Dear reviewer
Sorry for having forgotten to modify Table 3 and for having uploaded an unpolished version of the supplementary experiments. These mistakes are now corrected.